# Improved implementation of aspirin in pregnancy among Dutch gynecologists: Surveys in 2016 and 2021

Jeske Milou bij de Weg[1,2]*, Laura Visser[1,2], Martijn Alexander Oudijk[1,2], Johanna Inge Petra de Vries[1,2], Christianne Johanna Maria de Groot[1,2], Marjon Alina de Boer[1,2]

1 Amsterdam UMC location Vrije Universiteit Amsterdam, Obstetrics and Gynecology, Amsterdam, Netherlands, 2 Amsterdam Reproduction and Development Research Institute, Pregnancy and Birth, Amsterdam, Netherlands

* j.bijdeweg@amsterdamumc.nl

## Abstract

### Objective

To evaluate the implementation of low-dose aspirin in pregnancy for the prevention of utero-placental complications among gynecologists in the Netherlands between 2016 and 2021. In this timeframe, a national guideline about aspirin in pregnancy was introduced by the Dutch Society of Obstetrics and Gynecology.

### Materials and methods

A national online survey among Dutch gynecologists and residents was performed. An online questionnaire was distributed among the members of the Dutch Society of Obstetrics and Gynecology in April 2016 and April 2021. Main outcome measure was the proportion of gynecologists indicating prescription of aspirin in pregnancy for high and moderate risk indications.

### Results

In 2016, 133 respondents completed the survey, and in 2021 231. For all indications mentioned in the guideline there was an increase in prescribing aspirin in 2021 in comparison to 2016. More specifically, the percentage of gynecologists prescribing aspirin for a history of preeclampsia before 34 weeks, between 34 and 37 weeks and at term increased from respectively 94% to 100%, 39% to 98%, and 15% to 97%. Consultant obstetricians and respondents working in an university hospital did not more often indicate the prescription of aspirin for tertiary care indications in 2021. Future use of a prediction model was suggested in the narrative comments.

### Conclusion

Implementation of aspirin in pregnancy among Dutch gynecologists substantially improved after a five year timeframe in which the national guideline on aspirin during pregnancy was introduced and trials confirming the effect of aspirin were published.

**Data Availability Statement:** All relevant data are within the article and its Supporting Information files.

**Funding:** The author(s) received no specific funding for this work.

**Competing interests:** The authors have declared that no competing interests exist.

## Introduction

The efficacy of low-dose aspirin for the prevention of obstetrical complications has been an important research question for several decades [1]. Numerous studies have determined the effect of low-dose aspirin on the incidence of preeclampsia (PE) and other obstetric complications including fetal growth restriction (FGR) and preterm birth, with conflicting results [2–8]. In 2010, a meta-analysis showed that early initiation with aspirin before sixteen weeks of gestation resulted in risk reduction of obstetrical complications, in contrast to a later start after sixteen weeks of gestation [9]. After this meta-analysis in 2010, international guidelines had to be created for the translation to clinical practice [10]. It took three to eight years before international guidelines were published [11–14]. Initially, additions to the guidelines concerning hypertensive disorders of pregnancy (HDP) were made. Later, separate modules and/or guidelines on aspirin in pregnancy were created. In the Netherlands, aspirin as risk reducing therapy was added to the national guideline on HDP in 2018 [15]. In 2019, a separate module about aspirin in pregnancy was published [16].

The next step in implementation is the use of guidelines in clinical practice. Evidence based clinical guidelines are sometimes unmanageable because of the high volume of evidence and the poor translation to clinical practice [17]. Limited use of guidelines hinders reduction of adverse outcomes due to suboptimal treatment or prevention [18]. Multiple factors, so-called barriers in the implementation literature, are described as explanation for the challenge of implementation of guidelines [19]. These barriers include for instance characteristics of the caregiver and patient, and methods of information distribution.

We investigated the implementation in clinical practice over time. We hypothesized that the implementation of aspirin would have improved over time, for instance as a result of the introduction of the national guideline, module, and several published trials and meta-analyses confirming the effect of aspirin [15, 16, 20–26]. Therefore we performed a survey using questionnaires among gynecologists and residents on indications for which they prescribed aspirin in pregnancy in 2016 and repeated the questionnaire in 2021.

## Materials and methods

In April 2016 and April 2021 a survey was distributed among members of the Dutch Society of Obstetrics and Gynecology. In 2016, the survey was sent to the 979 gynecologists registered at the Dutch Society of Obstetrics and Gynecology. In 2021, the survey was distributed among the 1511 registered gynecologists and residents, since we wished to explore the implementation among residents as well. Hereafter, gynecologists and residents will be mentioned as gynecologists. Members were invited by e-mail to conduct a questionnaire through a provided link. In 2016, the questionnaire was designed in Google forms. In 2021, the questionnaire was designed in Castor Electronic Data Capture [27] due to stricter rules on privacy. The study was approved by the medical ethical committee of the Amsterdam UMC location VUmc (2020.0712). No informed consent was obtained, since for an anonymous survey it is not required. The survey consisted of selected responses and open questions. An overview of the survey is available in **S1 Table.** Respondents were also invited to provide narrative suggestions to prescribe aspirin for other indications than mentioned in the survey. No personal identifying information was collected to maintain confidentially. Participation was voluntary and no incentives for participating were provided. Gynecologists could participate in 2016 and 2021.

### Statistical analyses

Descriptive statistics were performed to present the raw data of the survey. Median with interquartile ranges (IQR) were reported for non-normally distributed variables and proportions were reported for categorical variables. To analyze the difference between 2016 and 2021, measurements

were considered unpaired since it was unknown if and which percentage of the respondents answered the questionnaire both in 2016 and 2021. Chi Square Test or Fisher's Exact Test were performed for categorical variables, and Mann Whitney U Test for numerical variables. In addition, we performed subgroup analyses investigating the influence of being a consultant obstetrician or working in an university hospital in prescribing aspirin for tertiary care indications (SLE, APS, other auto immune diseases and kidney disease). We expected that consultant obstetricians and respondents working in an university hospital more often prescribe aspirin for the tertiary care indications, because of higher awareness elicited by a population at risk for more severe illness. For the categorical variable of type of hospital, a dummy variable was created. The associations between the respondents characteristics and the above mentioned indications to prescribe aspirin were analyzed using logistic regression. SPSS version 26.0 (SPSS Inc., Chicago, IL, USA) was used to perform the statistical analyses. Results were considered significant at the two-sided 5% level.

## Results

### Respondents

In 2016, 133 out of the 979 invited gynecologists completed the survey, resulting in a response rate of 13.6%. In 2021, 231 out of the 1511 invited gynecologists completed the survey, resulting in a response rate of 15.3%. Characteristics revealed that in 2016, almost a quarter (24.1%) of the respondents reported to be a consultant obstetrician, and in 2021 18.1%. Characteristics of the respondents are depicted in **Table 1**.

### Indications

A significant rise over time in the percentage of gynecologists indicating prescription of aspirin for all indications described in the guideline, except for APS for which the percentage was already high in 2016, is illustrated **in Fig 1**. In the **S2 Table** an overview of the results of all questions is given.

### Prescription

Details about the advices on the dose, time of administering, gestational age of start and stop of aspirin are depicted in **Table 2**.

**Table 1. Characteristics of the responding gynecologists and residents in 2016 and 2021.**

|  | 2016 | 2021 | p-value |
|---|---|---|---|
|  | n = 133 | n = 231 |  |
| **Years of practice** |  |  | p = 0.034 |
| < 5 years | 34 (25.6) | 90 (39.0) |  |
| 5–15 years | 63 (47.4) | 88 (38.1) |  |
| > 15 years | 36 (27.1) | 53 (22.9) |  |
| **Type of hospital** |  |  | p = 0.705 |
| University | 25 (18.8) | 44 (19.0) |  |
| Teaching | 67 (50.4) | 125 (54.1) |  |
| Non-teaching | 41 (30.8) | 62 (26.8) |  |
| **Consultant obstetricians** | 32 (24.1) | 42 (18.2) | p = 0.180 |
| **Perinatal care in half-days per week** |  |  | p = 0.008 |
| When on-duty | 5 (3.8) | 28 (12.1) |  |
| 1–4 | 60 (45.1) | 113 (48.9) |  |
| 5 or more | 68 (51.1) | 90 (39.0) |  |

Data are depicted as number (%).

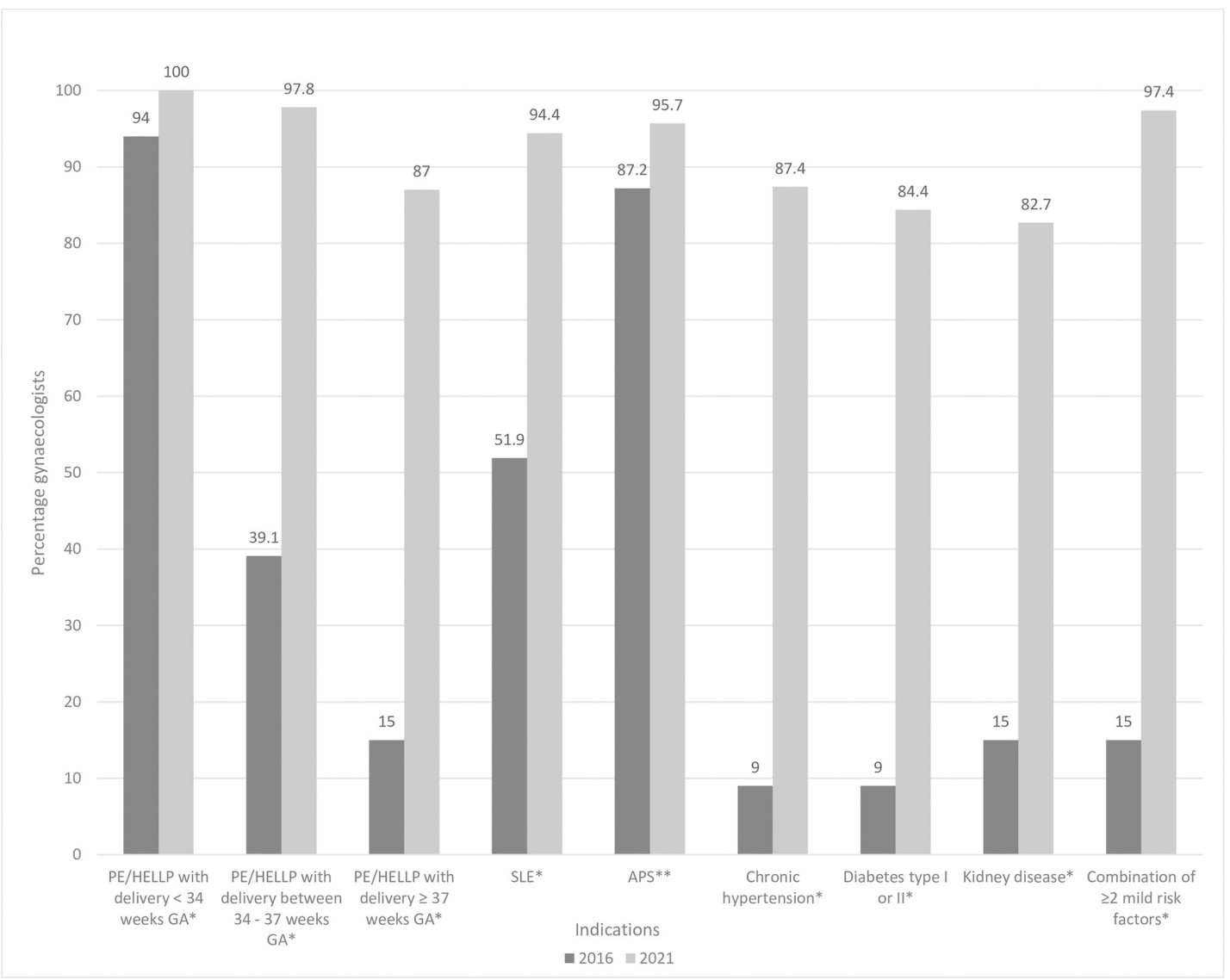

**Fig 1. Gynecologists and residents indicating prescription of aspirin during pregnancy: Comparison of 2016 and 2021.** PE, preeclampsia; HELLP, hemolysis elevated liver enzymes low platelets; GA, gestational age; SLE, systemic lupus erythematosus; APS, antiphospholipid syndrome. * Statistically significant; p = 0.000. **Statistically not significant; p = 0.250.

## Subgroup analyses

Being a consultant obstetrician or working in an university or teaching hospital was not significantly related to prescribing aspirin for the tertiary care indications SLE, APS, other autoimmune diseases and kidney disease, see Table 3.

## Narrative suggestions

In 2021, twelve gynecologists responded to advice aspirin for other risk populations. Five respondents used a prediction tool for setting the indication for aspirin use during pregnancy. The other suggestions were: unexplained stillbirths (n = 2), smoking cigarettes or using cocaine (n = 1), atherosclerosis such as a history of myocardial infarction or transient ischemic

**Table 2. Gynecologists' advices on prescription of aspirin in 2016 and 2021.**

| | 2016 | 2021 | p-value |
|---|---|---|---|
| | n = 133 | n = 231 | |
| **Dose of aspirin** | | | p = 0.000 |
| <80 mg | 4 (3.0) | 0 (0) | |
| 80 – 100mg | 129 (97.0) | 198 (85.7) | |
| 150 – 160mg | 0 (0) | 33 (14.3) | |
| **Time of administering aspirin** | | | p = 0.000 |
| In the evening | 27 (20.3) | 167 (72.3) | |
| In the morning | 14 (10.5) | 12 (5.2) | |
| At a fixed time during the day | 92 (69.2) | 52 (22.5) | |
| **Start of aspirin at GA in weeks** | 6.5 [6.0–6.5] | 12.0 [10.0–12.0] | p = 0.000 |
| **Stop of aspirin at GA in weeks** | 36.0 [36.0–36.0] | 36.0 [36.0–36.0] | p = 0.329 |
| **Earlier stop of aspirin in case of** | | | |
| Vaginal blood loss | 47 (35.3) | 78 (33.8) | p = 0.761 |
| Imminent preterm labor | 26 (19.5) | 107 (46.3) | p = 0.000 |
| Stomach complaints | 31 (23.3) | 127 (55.0) | p = 0.000 |
| Other | 12 (9.1) | 36 (15.6) | p = 0.079 |

Data are depicted as number (%) or median [IQR]. Chi-square Test is performed on the data depicted as number (%) and Mann-Whitney U Test by data depicted as median [IQR].

GA, gestational age.

attack (n = 1), other auto-immune diseases (n = 1), fetal growth restriction with or without histologically confirmed utero-placental insufficiency (n = 2).

## Discussion

### Main findings

This national survey among gynecologists revealed a significant rise of gynecologists indicating prescription of aspirin for all moderate and high risk populations. Consultant obstetricians and respondents working in an university hospital did not more often indicate the prescription of aspirin for tertiary care indications.

### Interpretation in the light of other evidence

A significant rise in indicating prescription of aspirin over time was seen in the group with a history of HDP, with the highest indication rates for preterm PE or hemolysis elevated liver

**Table 3. Subgroup analyses of respondents characteristics to tertiary care aspirin indications in 2021.**

| | SLE | APS | Other auto-immune disease | Kidney disease |
|---|---|---|---|---|
| **Consultant obstetrician** | OR 2.780 (95% CI 0.351–21.986; p = 0.333) | N/A* | OR 1.997 (95% CI 0.994–4.014; p = 0.052) | OR 3.164 (95% CI 0.927–10.805; p = 0.066) |
| **University hospital** | OR 3.772 (95% CI 0.425–33.442; p = 0.233) | OR 4.607 (95% CI 0.535–39.709; p = 0.165) | OR 2.155 (95% CI 0.888–5.229; p = 0.089) | OR 1.134 (95% CI 0.441–2.915; p = 0.794) |
| **Teaching-hospital** | OR 1.479 (95% CI 0.450–4.862; p = 0.520) | OR 4.357 (95% CI 1.052–18.054; p = 0.042) | OR 1.820 (95% CI 0.871–3.801; p = 0.111) | OR 1.853 (95% CI 0.845–4.062; p = 0.123) |

SLE, systemic lupus erythematosus; APS, antiphospholipid syndrome; OR, odds ratio; CI, confidence interval. *Not applicable because of zero in the 2x2 table (zero consultant obstetricians responded 'no' in prescribing aspirin for the indication APS).

enzymes low platelets (HELLP) syndrome. This can probably be explained by the higher recurrence risk in case of preterm PE and the stronger risk reducing effect of aspirin on preterm PE in comparison to term PE [23, 28]. Moreover, aspirin as risk-reducing therapy in this population is already implemented in the Dutch clinical practice since the publications of Dekker et al in 1993 and Wallenburg in 1995 [29, 30]. The rise in indicating to prescribe aspirin for auto-immune diseases was statistically significant, except for APS. The most likely explanation is that the awareness of APS was already high in 2016 and further increased in 2021. Since 1997, there was already evidence on the beneficial effect of aspirin and low-molecular weight heparin on live births in pregnant women with APS [31]. Also in the group of maternal diseases a significant rise over time was shown. In a systematic review and meta-analysis of 2016 in which risk factors for developing PE were investigated, chronic hypertension showed a pooled relative risk of developing PE of 5.1 (4.0–6.5), pregestational diabetes of 3.7 (3.1–4.3) and chronic kidney disease of 1.8 (1.5–2.0) [32]. More insight in the elevated risk of women with maternal diseases probably have led to the increased prescription rates. At last, in the group of moderate risk factors also higher indication rates over time were seen, especially for a combination of ≥2 moderate risk factors. In the previous mentioned systematic review and meta-analysis of Bartsch et al, risk factors such as nulliparity, obesity, advanced maternal age and multifetal pregnancy showed an elevated pooled risk, suggesting that women with moderate risk factors may benefit from low-dose aspirin as well [32]. Screening only on maternal factors is a simplified risk calculation when compared to other screening methods in which in addition for instance Doppler measurements and serum placental growth factor and pregnancy-associated plasma protein A are determined [33, 34].

Around 10% of the gynecologists responded to advice aspirin in case of recurrent miscarriages, despite the fact that a systematic review revealed no evidence of a beneficial effect of low-dose aspirin in this group [35]. We did not investigate the motivation for this advice, but one could speculate that the methods to distribute the result of this systematic review have not been sufficient to reach some gynecologists, a known barrier of the implementation of new results.

In 2021, a small proportion of the respondents indicate prescription of aspirin to women with a history of spontaneous preterm birth <34 weeks of gestation. The evidence on the risk reducing effect of aspirin on spontaneous preterm birth varies [9, 36–40]. In addition, there is some evidence that aspirin may reduce preterm birth in nulliparous women without risk factors for hypertensive disorders of pregnancy [41]. If future studies can confirm the risk reducing effect in nulliparous women or women with a history of preterm birth, these indications may be added to the list of indications for aspirin during pregnancy.

The most frequently prescribed aspirin dose is stable over time, namely 80-100mg. Although in 2021, 14.3% of the respondents advised higher aspirin doses of 150-160mg. This might be explained by recent meta-analyses suggesting that aspirin doses above 100 mg may be more effective than lower doses [9, 23, 42]. However, this evidence is based on indirect comparisons. In 2021, the advice to take aspirin in the evening was considerably enhanced. This advice is based on the circadian rhythm of new platelets where the peak release is in the late night and early morning [43]. Since aspirin is quite rapidly absorbed with a time to maximal plasma level of 30 minutes to two hours, evening intake can inhibit this peak. In the cardiovascular population, evening intake is associated with a more stable platelet inhibition over 24 hours in comparison to morning intake [44]. Awareness of this pharmacologically explanation might stimulate further implementation and should therefore be emphasized in education about this topic.

Subgroup analyses showed that consultant obstetricians and respondents working in an university hospital did not more often indicate the prescription of aspirin for tertiary care

indications. These promising results show the general awareness among gynecologists and residents among all type of hospitals. Although, awareness among clinical caregivers is no guarantee for awareness among and adherence in patients.

The improved implementation might be explained by the introduction of the national guideline in 2018 and module in 2019 [15, 16]. In addition, the Dutch Society of Obstetrics and Gynecology strives for improvement of implementation and overcoming barriers in the methods of information distribution. Therefore, in 2020 it published a document in Dutch with advice on optimizing the quality circle: the process from collecting evidence by trials to implementation in clinical practice [45]. In case of publication of a new guideline, the Dutch Society of Obstetrics and Gynecology communicates this by notifications to their members. As well as presenting updates in their newsletter, promoting discussions during congresses and other meetings, and/or teaching by e-learnings. These procedures probably have contributed to the improved implementation of aspirin in clinical practice. Moreover, awareness has been emphasized by large RCT's and meta-analyses over the last years, confirming the risk-reducing effect of aspirin during pregnancy [20–26]. The studies probably created more assurance of the efficacy of aspirin and confirmed the lack of adverse events. The latter being specifically important when prescribing aspirin to women with a moderate risk profile.

## Future

We showed that implementation of new therapy in clinical practice remains a challenge. Pathman et al described the process of implementation of clinical guidelines in the following four steps: 1) Awareness, 2) Agreement, 3) Adoption, and 4) Adherence [46]. In our study, we tested the adherence of the caregiver to the clinical guideline, in the study of Pathman et al described as 'actually succeeding in following the guideline at appropriate times'. Research on implementation of new therapies and clinical guidelines should be more frequently be performed, since unsuccessful implementation could lead to harm of the patients. Our study shows that clinical caregivers are willing to expose to implementation testing, since a substantial part of the invited gynecologists and residents responded to the survey.

The use of a prediction model might be helpful in the successful implementation of aspirin in clinical practice. In the narrative suggestions, five gynecologists plead for the use of a prediction model to set the indication for aspirin during pregnancy. The Fetal Medicine Foundation provides a biomarkers, ultrasound and history based prediction tool to identify women at high-risk for preterm PE [47]. Such a prediction method was tested in the ASPRE trial, detecting 76% of the preterm PE cases [34]. A systematic review to prediction models for PE in 2019 concluded that there is a large variety of prediction models with great heterogeneity among the study methods [48]. In addition, most prediction tools were not validated. Therefore, no advice on the best performing prediction model could be given. Before implementing in clinical practice, model validation should be performed [48].

After adherence of the caregiver comes adherence of the patient. In 2020, we showed that the awareness about aspirin as risk-reducing therapy on HDP in women at risk was only present in 51.9% [49]. Low awareness could result in non-adherence. Previous research reported 21–46% of aspirin non-adherence during pregnancy [50]. In the ASPRE trial, a placebo-controlled trial studying the risk-reducing effect of aspirin on preterm PE, showed that in women with aspirin adherence ≥90%, aspirin had a higher risk-reducing effect in comparison to women with aspirin adherence <90% (OR 0.24; 95% CI 0.09–0.65 versus OR 0.59; 95% CI 0.23–1.53) [22]. Research to optimize aspirin adherence during pregnancy should be performed.

## Strengths and limitations

As far as we know, we were the first who investigated the implementation of aspirin during pregnancy by gynecologists. The five-year evaluation on the change in implementation over time is an important strength of our study. By performing this survey, we completed the quality circle [45]. In addition, we investigated a homogeneous population with access to the same national guideline which is representative for the Dutch group of gynecologists. A limitation of our study is the relatively low response rates of 13.6% and 15.3% in 2016 and 2021, respectively. Secondly, a relatively large proportion of the respondents worked in an university hospital, which might give an overestimation of the implementation. At last, our survey might be not representative for all countries, although the attached survey in S1 Table facilitates reproduction of this study in other countries.

## Conclusion

In conclusion, the implementation of aspirin in pregnancy for women at moderate or high risk among Dutch gynecologists is improved over time. The improved implementation can probably be explained by the introduction of the national guideline and published trials confirming the risk-reducing effect of aspirin. The next step in implementation is patients' adherence. Future studies should focus on the optimization of screening tools for obstetrical complications and improvement of aspirin adherence of pregnant women.

## Supporting information

**S1 Table. Overview of the questions of the survey.** HDP, hypertensive disorders of pregnancy; PE, preeclampsia; HELLP, hemolysis elevated liver enzymes low platelets; GA, gestational age; PIH, pregnancy induced hypertension; FGR, fetal growth restriction; SPTB, spontaneous preterm birth; SLE, systemic lupus erythematosus; APS, antiphospholipid syndrome; BMI, body mass index.
(DOCX)

**S2 Table. Gynecologists and residents indicating prescription of aspirin during pregnancy: Comparison of 2016 and 2021.** Data are depicted as number (%). HDP, hypertensive disorders of pregnancy; PE, preeclampsia; HELLP, hemolysis elevated liver enzymes low platelets; GA, gestational age; PIH, pregnancy induced hypertension; FGR, fetal growth restriction; SPTB, spontaneous preterm birth; SLE, systemic lupus erythematosus; APS, antiphospholipid syndrome; BMI, body mass index; N/A, not available.
(TIF)

## Acknowledgments

We want to thank the Dutch Society of Obstetrics and Gynecology for their help with collecting our data by spreading the survey among their members. We also thank N. Schuster of the department of epidemiology and data science for her advice and help in performing the statistical analyses.

## Author Contributions

**Conceptualization:** Martijn Alexander Oudijk, Christianne Johanna Maria de Groot, Marjon Alina de Boer.

**Data curation:** Jeske Milou bij de Weg, Laura Visser.

**Formal analysis:** Jeske Milou bij de Weg.

**Project administration:** Jeske Milou bij de Weg.

**Supervision:** Martijn Alexander Oudijk, Johanna Inge Petra de Vries, Christianne Johanna Maria de Groot, Marjon Alina de Boer.

**Writing – original draft:** Jeske Milou bij de Weg.

**Writing – review & editing:** Laura Visser, Martijn Alexander Oudijk, Johanna Inge Petra de Vries, Christianne Johanna Maria de Groot, Marjon Alina de Boer.

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
