## [Decision Letter · Decision Letter 0]

22 Mar 2022

PONE-D-22-03768Improved implementation of aspirin in pregnancy among Dutch gynecologists: surveys in 2016 and 2021.PLOS ONE

Dear Dr. bij de Weg,

Thank you for submitting your manuscript to PLOS ONE. After careful consideration, we feel that it has merit but does not fully meet PLOS ONE’s publication criteria as it currently stands. Therefore, we invite you to submit a revised version of the manuscript that addresses the points raised during the review process.

We look forward to receiving your revised manuscript.

Kind regards,

Sara Ornaghi, M.D., Ph.D.

Academic Editor

PLOS ONE

Journal Requirements:

Reviewers' comments:

Reviewer's Responses to Questions

**Comments to the Author**

1. Is the manuscript technically sound, and do the data support the conclusions?

Reviewer #1: Yes

Reviewer #2: Yes

2. Has the statistical analysis been performed appropriately and rigorously? 

Reviewer #1: Yes

Reviewer #2: Yes

3. Have the authors made all data underlying the findings in their manuscript fully available?

Reviewer #1: Yes

Reviewer #2: Yes

4. Is the manuscript presented in an intelligible fashion and written in standard English?

Reviewer #1: Yes

Reviewer #2: Yes

5. Review Comments to the Author

Reviewer #1: I found this research on the improved implementation of aspirin in pregnancy among Dutch gynecologists interesting.

The study deals with a highly topical obstetric issue yet to be defined at international level and shows the practical difficulties in implementing a guideline.

The data are clearly presented and adequately support the conclusions and limitations.

The major criticism is that adherence to the research involved a low percentage of gynecologists 13,6 % in 2016 and 15,3 % in 2021.

The recipients of the questionnaire is unclear in some parts of the manuscript - gynecologists (in training)???

Reviewer #2: The author sought to perform a survey on the implementation of the use of aspirin in preventing preeclampsia after the introduction of national guidelines in the Netherlands comparing 2016 and 2020.

This is a very nice example of National improvement of health practice.

One limitation, well described by the authors, is the tertiary care background of those who partecipated.

I would suggest to the authors to add in the “limitations” also the low rate of responders: around 13%, which is quite low.

The authors presented a detailed description of the change in timing of the administration (evening versus other timings) and in the prescription of the dosage (low such as 80-100 mg versus high dosage as described in the ASPRE trial 150 mg).

I suggest to the Authors to perform in the next years a comparison between those gynecologists who rely on the anamnestic screening of preeclampsia (as stated by the Dutch National guidelines) compared to the bio markers+ultrasound+history based screening proposed by the FMF and used in the ASPRE trial. I suggest to include this topic in the Conclusions as future research.

6. PLOS authors have the option to publish the peer review history of their article (what does this mean?). If published, this will include your full peer review and any attached files.

Reviewer #1: No

Reviewer #2: **Yes: **Annalisa Inversetti

---

## [Author Response · Author response to Decision Letter 0]

27 Apr 2022

I refer to the uploaded rebuttal letter for our response to the editor and reviewers.

---

## [Editor Report · Decision Letter 1]

5 May 2022

Improved implementation of aspirin in pregnancy among Dutch gynecologists: surveys in 2016 and 2021.

PONE-D-22-03768R1

Dear Dr. bij de Weg,

We’re pleased to inform you that your manuscript has been judged scientifically suitable for publication and will be formally accepted for publication once it meets all outstanding technical requirements.

Kind regards,

Sara Ornaghi, M.D., Ph.D.

Academic Editor

PLOS ONE
---

## [Editor Report · Acceptance letter]

31 May 2022

PONE-D-22-03768R1 

Improved implementation of aspirin in pregnancy among Dutch gynecologists: surveys in 2016 and 2021. 

Dear Dr. bij de Weg:

I'm pleased to inform you that your manuscript has been deemed suitable for publication in PLOS ONE. Congratulations! Your manuscript is now with our production department. 

Kind regards, 

on behalf of

Dr. Sara Ornaghi 

Academic Editor

PLOS ONE